# Anatomically-Enhanced URO dot AI: Multi-Stage Fine-Tuning of Foundation Models for Precise Urinary Stone Segmentation

**Byeongin Moon**[*1]                                                        BLUEIN@AIDOT.AI
**Jaehoon Oh**[*2]                                                          OJJAI@HANYANG.AC.KR
**Young Eun Yoon**[*3]                                          UROLOGISTYOON@HANYANG.AC.KR
**Dong Keon Lee**[4]                                                   STOLENEGG@GMAIL.COM
**Gihun Lee**[1]                                                         1007HOON@AIDOT.AI
**Mingyeong Park**[1]                                               CYZLALSRUD96@AIDOT.AI
**Heesun Choi**[1]                                                        HEESUN@AIDOT.AI
**Hansol Choi**[†1]                                                       SOLCHOI@AIDOT.AI
**Jaehoon Jeong**[†1]                                                      JMAN@AIDOT.AI

[1] *AIDOT Inc., Seoul, Republic of Korea*

[2] *Department of Emergency Medicine, College of Medicine, Hanyang University, Seoul, Republic of Korea*

[3] *Department of Urology, College of Medicine, Hanyang University, Seoul, Republic of Korea*

[4] *Department of Emergency Medicine, College of Medicine, Seoul National University Bundang Hospital, Seongnam, Republic of Korea*

## Abstract

Automated segmentation of urinary stones in non-contrast CT (NCCT) is challenging due to small lesion size and voxel sparsity. We propose a hierarchical fine-tuning strategy based on the VISTA3D foundation model. Unlike the baseline approach that directly learns stone features from the entire volume (Stone FT only), our method first performs organ-aware fine-tuning to learn the urinary tract's anatomy, subsequently transferring these weights for stone-specific segmentation. On 119 test cases, the method achieved 95.69% stone-level and 96.64% patient-level sensitivity, demonstrating improved detection performance through anatomical context.

**Keywords:** Urinary Stone, Foundation Model, Hierarchical Fine-Tuning, VISTA3D.

## 1. Introduction

Accurate localization of urinary stones is essential for clinical intervention. While research primarily focuses on kidney stones (Li et al., 2022; Cui et al., 2021; Elton et al., 2022), automated detection of ureteral and bladder stones remains challenging due to the ureter's tortuous anatomy and a scarcity of annotated datasets. Furthermore, conventional statistical probability maps struggle to generalize across diverse patient anatomies (Längkvist et al., 2018).

To address this, we propose URO dot AI, leveraging the VISTA3D (Liu et al., 2024) foundation model pre-trained on large-scale medical data. Building on our previous work

---

[*] Contributed equally

[†] Corresponding author

on organ segmentation (Jang et al., 2024), we first establish explicit anatomical priors to define the boundaries of the urinary tract. This hierarchical approach effectively restricts the search space, isolating micro-lesions within the Kidney-Ureter-Bladder (KUB) region. Consequently, our framework improves stone detection across the KUB regions, increasing sensitivity for renal stones while addressing common false positives caused by the complex morphology of the ureter.

## 2. Methodology

### 2.1. Hierarchical Learning Framework Overview

The proposed pipeline (Figure 1) consists of ROI localization via UNETR (Hatamizadeh et al., 2022) followed by two distinct fine-tuning stages. We define "Hierarchical Fine-tuning" as a process where the model first learns the global morphology of the urinary tract (Stage 2A) before refining its weights to detect sparse stone voxels (Stage 2B).

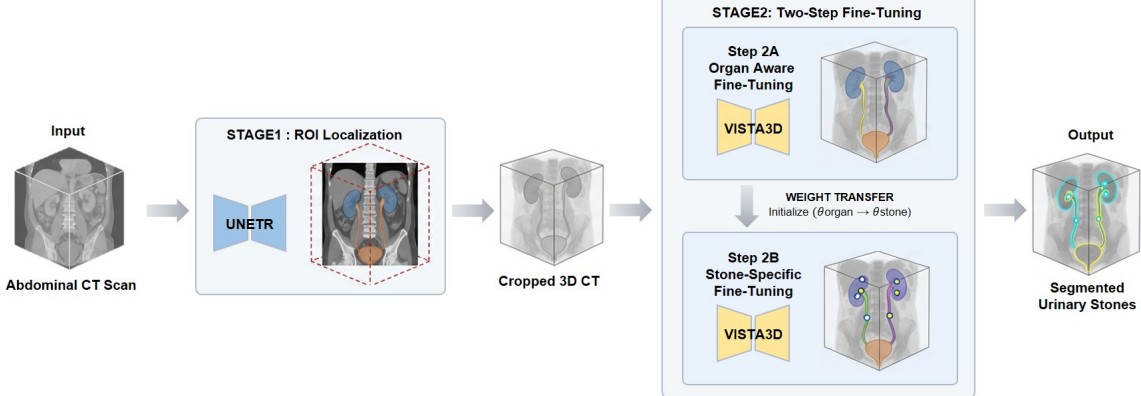

Figure 1: Overview of the proposed multi-stage hierarchical fine-tuning framework.

### 2.2. Stage 1: ROI Localization and Cropping (UNETR)

To optimize memory efficiency and narrow the search space, UNETR generates coarse kidney and bladder masks. These define a 3D ROI crop that encompasses the entire urinary trajectory while discarding irrelevant anatomical regions.

### 2.3. Stage 2: Two-Step Fine-Tuning Strategy (VISTA3D)

To evaluate the impact of anatomical priors, we compare two distinct training strategies:

- **Stone FT only (Baseline):** The VISTA3D backbone is fine-tuned directly on stone labels using ROI-cropped volumes. This baseline learns stone features without explicit organ context, reflecting a traditional single-stage approach.

- **Hierarchical Fine-tuning (Proposed):**

– **Step 2A (Organ-Aware):** The model is first specialized to segment nine urological structures (KUB) to internalize the anatomical boundaries where stones are clinically expected to reside.

– **Step 2B (Stone-Specific):** Step 2A weights serve as initialization, providing implicit spatial constraints that guide the model toward high-density voxels within the learned anatomical context.

## 2.4. Datasets

All datasets utilized in this study consist of non-contrast CT (NCCT) scans. The Organ Dataset (448 train / 72 val / 56 test) and Stone Dataset (809 train / 219 val / 119 test) were employed for organ and stone segmentation, respectively.

## 3. Results

The performance of our hierarchical approach (Table 1) significantly surpassed the baseline, boosting Stone-PPV from 44.37% to 60.79% while maintaining 95.69% sensitivity. This 16.42% improvement shows that organ-aware priors suppress false positives from extra-urinary high-intensity structures via organ-specific spatial weights. Regional analysis (Table 2) further highlights 100% sensitivity in proximal ureters and the bladder, validating the framework's ability to resolve diagnostic ambiguities in 3D volumes.

Table 1: Performance summary of the proposed method.

| Method | Patient-level Metrics | | | Stone-level Metrics | |
|---|---|---|---|---|---|
| | Sensitivity (Strict) | Sensitivity (Non-Strict) | Mean PPV | Sensitivity | PPV |
| Stone FT only (baseline) | 87.39% | 95.80% | 56.16% | 92.34% | 44.37% |
| Organ & Stone FT (proposed) | 93.28% | 96.64% | 72.37% | 95.69% | 60.79% |

*Note.* FT: Fine-tuning; Strict: all stones detected; Non-strict: at least one stone detected per patient.

Table 2: Stone-level sensitivity across different anatomical regions.

| Method | R-Kid | L-Kid | R-Prox | L-Prox | R-Mid | L-Mid | R-Dist | L-Dist | Bladder |
|---|---|---|---|---|---|---|---|---|---|
| Stone FT only (baseline) | 90.38% | 90.24% | 97.37% | 100.00% | 100.00% | 50.00% | 80.00% | 100.00% | 100.00% |
| Organ & Stone FT (proposed) | 96.15% | 92.68% | 100.00% | 100.00% | 83.33% | 75.00% | 95.00% | 88.89% | 100.00% |

*Note.* FT: Fine-tuning; R/L: Right/Left; Kid: Kidney; Prox/Mid/Dist: Proximal/Mid/Distal Ureter.

## 4. Conclusion

We demonstrate that hierarchical anatomical priors in a 3D foundation model significantly enhance microscopic stone detection by constraining the search space and suppressing false positives, even under data sparsity. This framework provides an effective strategy for anatomy-guided stone detection, successfully bridging large-scale foundation models with specialized clinical tasks.

## Acknowledgments

This work was conducted by AIDOT Inc. The authors are grateful to Hanyang University Hospital and Seoul National University Bundang Hospital for their essential support in providing the clinical datasets, which were accessed and used under the approval of their respective Institutional Review Boards (IRBs).

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
