# OpenReview forum: "Anatomically-Enhanced URO dot AI: Multi-Stage Fine-Tuning of Foundation Models for Precise Urinary Stone Segmentation"
_MIDL.io/2026/Short_Papers — MIDL 2026 - Short Papers Poster_

### Official Review · Reviewer_tHwH · 2026-05-04
**Show moreShow lessAn anatomically guided fine-tuning approach for urinary stone detection in CT with improved sensitivity and PPV but limited evaluation.**

**Rating:** 3
**Confidence:** 4

**Review:**

- The paper addresses a clinically relevant problem and proposes a reasonable idea: using anatomical priors to improve detection of small, sparse lesions.
-	The approach is potentially interesting and transferable to other tasks where lesions are small and spatially constrained.

However, several aspects are unclear:
- The baseline “Stone FT only” is not sufficiently defined, making the comparison hard to interpret.
- The claim that organ context is used as a spatial constraint is not explained; it appears more like sequential fine-tuning rather than an explicit constraint during inference.
- The “hierarchical” nature seems limited to staged training and ROI cropping.
- The evaluation is limited:
  - Only sensitivity and PPV are reported, while the task is framed as segmentation.
  - No segmentation metrics (e.g., Dice) are provided, but the title claims ‘precise segmentation’

- The false-positive burden remains substantial (PPV ≈ 60%), yet is not analyzed further.
- It is unclear whether stone-negative cases or relevant confounders (e.g., cysts, tumors, calcifications) are included.
- The claim of a “robust, scalable solution for automated urological diagnostics” is therefore not sufficiently supported.
- Overall, the paper presents an interesting idea, but lacks clarity and sufficient evaluation to make the contribution convincing or easily transferable.

**Summary:**

This paper proposes an anatomically informed pipeline for urinary stone segmentation in non-contrast CT. The method first localizes a kidney–ureter–bladder region of interest using UNETR, then applies a two-step fine-tuning strategy based on VISTA3D: organ-aware fine-tuning followed by stone-specific fine-tuning. The authors report improved stone-level sensitivity and positive predictive value compared with a “stone fine-tuning only” baseline, suggesting that anatomical context may help reduce false positives in urinary stone detection.

**Strengths:**

- Clinically relevant and challenging task.
- Interesting idea of incorporating anatomical context for small-lesion detection.
- Improvement over a simple baseline.
 - Potentially transferable concept to other applications.

**Weaknesses:**

- Baseline not clearly defined.
- Unclear use of organ context as spatial constraint.
- “Hierarchical” concept not well justified.
- Limited evaluation (no segmentation metrics, no false-positive analysis).
- No analysis on negative cases or confounding structures.
- Claims about robustness and clinical applicability are too strong.

**Justification Of Rating:**

The idea is relevant and potentially interesting for the community.
However, missing methodological clarity and insufficient evaluation limit the strength of the contribution.

---

### Decision · Program_Chairs · 2026-05-08

Accept (Poster)